# FeDa4Fair: Client-Level Federated Datasets for Fairness Evaluation

## Abstract

Federated Learning (FL) enables collaborative model training across multiple clients without sharing clients' private data. However, the diverse and often conflicting biases present across clients pose significant challenges to model fairness. Current fairness-enhancing FL solutions often fall short, as they typically mitigate biases for a single, usually binary, sensitive attribute, while ignoring the heterogeneous fairness needs that exist in real-world settings. Moreover, these solutions often evaluate unfairness reduction only on the server side, hiding persistent unfairness at the individual client level. To support more robust and reproducible fairness research in FL, we introduce a comprehensive benchmarking framework for fairness-aware FL at both the global and client levels. Our contributions are three-fold: (1) We introduce FeDa4Fair, a library to create tabular datasets tailored to evaluating fair FL methods under heterogeneous client bias; (2) we release four bias-heterogeneous datasets and corresponding benchmarks to compare fairness mitigation methods in a controlled environment; (3) we provide ready-to-use functions for evaluating fairness outcomes for these datasets.

## 1 Introduction

With the increasing application of Machine Learning (ML) in all economic and societal sectors, the demand for its responsible use is becoming more prominent. This led to the introduction of several Artificial Intelligence (AI) regulations (Roberts et al., 2021; Biden, 2023; Commission, 2019; Madiega, 2021) and the emergence of new research fields such as explainability (Bodria et al., 2023), fairness (Caton & Haas, 2024), and user privacy (Liu et al., 2021).

One solution which is commonly adopted to mitigate users' privacy risk is Federated Learning (FL) (McMahan et al., 2017). FL enables a collaborative training of an ML model without requiring users, commonly called clients, to share their raw data. A significant challenge in FL is handling non-independent and identically distributed (non-i.i.d.) data across clients. Training an FL model in these settings degrades model utility and decreases its fairness guarantees. While progress has been made in addressing the utility degradation problem under non-i.i.d. conditions (Lu et al., 2024), efforts to mitigate the resulting unfairness remain limited. Existing fair FL approaches (Papadaki et al., 2022; Corbucci et al., 2024; Abay et al., 2020) typically aim to reduce unfairness for underrepresented groups using group-level metrics such as Demographic Disparity or Equalized Odds Difference (Barocas et al., 2017).

Notably, current FL methodologies operate under the simplified assumption that the bias distribution in terms of sensitive attributes and their most vulnerable groups is uniform across all clients. This assumption can lead to a dangerous illusion of fairness, neglecting the inherent heterogeneous bias distribution of real-world data, where clients may operate under differing political, cultural, or socioeconomic environments, legal and regulatory frameworks, data collection pipelines, or fairness objectives, leading to non-uniform biases. Consequently, having a single global model evaluated on the server using a single metric can result in a model that, while appearing globally fair, is unfair for a few individual clients or specific demographic groups due to unaddressed attributes. While some clients may benefit from the fairer model, others do not. This can lead to bias propagation across the federation (Chang & Shokri, 2023) and result in global models that are less fair and accurate than locally trained ones (Corbucci et al., 2025).

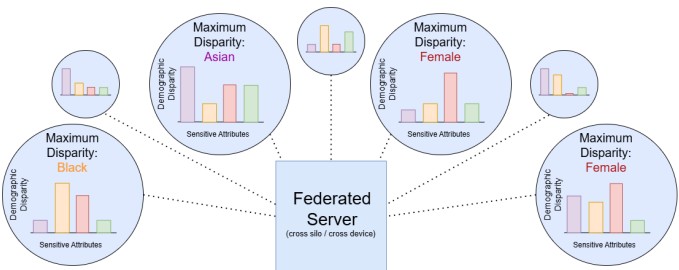

Figure 1: A pictorial representation of the FL scenarios tackled by FeDa4Fair. Clients exhibit varying levels of unfairness, here depicted as a high value of Demographic Disparity. FeDa4Fair creates data where fairness metrics reveal inequalities across attribute values (e.g., Black, Asian), across attributes (e.g., race vs. gender), or both.

In this work, we argue that a more granular client-level evaluation is necessary to address the current limitations. We identify two scenarios that current fair FL solutions do not capture: (I) *value bias*, where clients have data biased toward different *values* of the *same* sensitive attribute; and (II) *attribute bias*, where clients have data biased toward different sensitive *attributes* than those mitigated by the federation-level intervention. We visualize these scenarios in Figure 1. As an illustrative hypothetical example, value bias can arise in a race-fair facial recognition FL model trained on user photos. Since individuals typically possess more photos of themselves, an individual's local dataset is naturally biased toward their own race. When the federation consists of a majority (e.g., "Asians") and a minority group (e.g., "African-American"), the majority's data inevitably skews the fairness objective. This can create an illusion of fairness when the model is evaluated with a single global metric on the server, while it remains unfair for individual clients who receive a model that fails to meet their needs. As a result, conflicting client interests can make it difficult, if not impossible, to achieve meaningful fairness in the resulting federated model, even if a federation-level mitigation was implemented. Similarly, attribute bias may occur when clients' data are biased with respect to different attributes (e.g., gender vs. race). Such disparities complicate efforts to ensure fairness across multiple attributes in FL.

To fill this gap, we introduce FeDa4Fair, the first framework and library for creating benchmarking datasets tailored for the evaluation of fair FL methods in these realistic, bias-heterogeneous scenarios. By enabling systematic and reproducible experimentation, FeDa4Fair addresses the absence of standardized datasets available to researchers developing and evaluating fairness-aware FL methods. We summarize our contributions below:

(1) We introduce FeDa4Fair[1], a library for creating benchmarking datasets to assess fair FL approaches in diverse fairness scenarios, enabling comprehensive client-level fairness assessments.
(2) We release four benchmarking datasets incorporating heterogeneous bias distributions across clients for different FL settings.
(3) We provide results for two baseline unfair (non-mitigated) and fair (mitigated) FL methods on these four datasets, highlighting the limitations of current approaches and the need for a shift in the fair-FL evaluation pipeline.

## 2 BACKGROUND AND RELATED WORK

### 2.1 FEDERATED LEARNING

FL (McMahan et al., 2017) is a distributed training framework that enables $K$ clients to train a shared ML model without exposing their private datasets $D_k$. A central server $S$ orchestrates training by selecting a subset $\chi$ of available clients for $r \in [0, R]$ rounds and aggregating their model updates. At round $r = 0$, the server shares a model $\theta_0$ with random weights. The training procedure depends on the chosen aggregation method. A frequently used approach is Federated Averaging (FedAvg) (McMahan et al., 2017). In this case, clients update $\theta_r$ locally for $E$ epochs before sending updates to $S$, which aggregates them as $\theta_{r+1} \leftarrow \sum_{k=1}^{K} \frac{n_k}{n} \theta^k$, where $\theta^k$ are the parameters updated

---

[1] https://anonymous.4open.science/r/FeDa4Fair-AEB3/

locally by client $k$. FedAvg minimizes communication overhead while preserving performance, making it widely applied in FL. Based on the number and the availability of the clients involved in the training, we can distinguish between cross-silo and cross-device FL. In the cross-silo scenario, there are typically tens to hundreds of clients, such as hospitals or companies, that are always available during the training and possess large volumes of data. On the contrary, the cross-device scenario involves a larger number of clients, each holding a small number of data samples. In this case, the clients are only available under specific circumstances, e.g., a client could be a smartphone and could be available while charging. Several frameworks are available to simulate model training using FL (Riviera et al., 2023). One of the most well-known frameworks is Flower (Beutel et al., 2020), which we adopted for all the FL experiments presented in this paper. Additionally, the FeDa4Fair library we introduce is fully compatible with Flower, ensuring a seamless integration for researchers and practitioners.

## 2.2 FAIRNESS IN MACHINE LEARNING AND FEDERATED LEARNING

Fairness in ML refers to the principle that models' predictions should not systematically disadvantage individuals or groups based on sensitive attributes such as gender or race. To quantify how much an ML model is biased, multiple fairness metrics are available in the literature (Mehrabi et al., 2021; Caton & Haas, 2024). In this paper, we measure fairness using Demographic Disparity (DD) and Equalized Odds Difference (EOD) (Hardt et al., 2016). The former builds upon Demographic Parity (Barocas et al., 2017), which requires that the likelihood of a particular prediction outcome must not depend on the membership of a sensitive group. Formally, Demographic Parity can be expressed as: $\mathbb{P}(\hat{Y} = y \mid Z = z) = \mathbb{P}(\hat{Y} = y \mid Z \neq z)$ where $y$ is one of the possible targets predicted by the model and $z$ is one of the possible values of the sensitive attribute. DD then evaluates the maximum difference between the Demographic Parity of the different sensitive groups. Equalized odds difference instead demands equality of both the true positive and false positive rates across groups. Formally, it is defined as: $\mathbb{P}(\hat{Y} = y \mid Y = y, Z = z) - \mathbb{P}(\hat{Y} = y \mid Y = y, Z \neq z)$ for each $y \in Y$ and $z \in Z$.

Additionally, intersectional fairness captures forms of discrimination and societal effects that emerge from the intersection of multiple sensitive features (Crenshaw, 2013). For instance, young Black people may experience bias not because of age or race alone, but from their intersection. While each group may not be disadvantaged independently, their intersection can be. Addressing intersectional fairness is challenging, as these subgroups are often small and difficult to identify within the data.

Fairness is also a concern when training models using FL (Salazar et al., 2024; Vucinich & Zhu, 2023). In this context, it is not only important to ensure fairness across the different groups represented in the federation but also to understand the benefits individual clients gain from participating in training (Düsing & Cimiano, 2022; Yu et al., 2020). These benefits are usually measured in terms of model utility. However, prior work did not investigate participation benefits from a fairness perspective, despite proofs that FL is particularly vulnerable to bias propagation (Fontana et al., 2022; Chang & Shokri, 2023). These gaps are significant in federations where mitigation strategies reduce unfairness with respect to a specific sensitive attribute (Papadaki et al., 2022; Corbucci et al., 2024; Abay et al., 2020). In such cases, the choice of which sensitive attribute to mitigate can benefit the clients who are unfair toward that attribute while potentially damaging others who experience unfairness toward different attributes.

## 2.3 FEDERATED LEARNING DATASETS

Despite the growing popularity of FL research, the field lacks standardized benchmarking practices. This is particularly evident in two areas: the absence of a gold standard in terms of datasets that should be experimented on [2] and the inconsistent methodologies for data partitioning used to simulate realistic FL scenarios (Gutierrez et al., 2024b).

LEAF (Caldas et al., 2019) was the first attempt to establish a benchmarking dataset for FL. However, this benchmark only contains a dataset originally designed for the centralized context, which was then adapted to work for FL. Therefore, it does not effectively capture the different client-level distributions that could be present in an inherently federated dataset. To address this limitation,

---

[2]Federated Datasets in Research: https://flower.ai/blog/2024-12-02-federated-datasets-in-research/

researchers have proposed various approaches for simulating non-IID data distributions across clients (Gutierrez et al., 2024b). The use of the Dirichlet distribution emerged as a popular partitioning method to simulate the non-IID split. Solutions like NIID-Bench (Li et al., 2022) proposed a first benchmarking suite to compare different approaches and evaluate the impact on the model quality. More recently, FedArtML (Gutierrez et al., 2024a) proposed a similar solution to simulate and evaluate how data heterogeneity impacts the quality of the FL model.

A similar problem appears when fairness is taken into account during the FL process. Here, most of the datasets used in the literature are originally designed for a centralized context, and only contain a low number of samples, which makes training in a federated context difficult (Salazar et al., 2024). Beyond data heterogeneity concerns, to the best of our knowledge, no consensus exists on benchmark datasets or libraries designed to create data distributions with specific unfairness properties. As also pointed out by Taik et al. (2025), the existence of such datasets would benefit the evaluation of FL models, especially, when dealing with clients with varying fairness preferences and objectives. Not only researchers working on methods to mitigate model unfairness, but also those evaluating existing approaches in an FL environment, would profit from a standardized benchmarking dataset.

## 3 FEDA4FAIR

Meaningful comparison of fair FL methods requires a shift in the evaluation pipeline: from evaluating single global models on the server in simplified client settings to individual-level evaluation in diverse, bias-heterogeneous client settings. To facilitate this, we introduce "Client-Level **Fe**derated **Da**tasets **for Fair**ness Evaluation" (**FeDa4Fair**), a library designed to create datasets for this purpose and help researchers to investigate fairness across a wide range of FL scenarios. Unlike existing evaluation pipelines for FL models' fairness evaluation, FeDa4Fair provides data with a natural client-level split guaranteeing a non-i.i.d. distribution of the data and enabling systematic client-level fairness evaluation.

FeDa4Fair is built on top of the fairness-relevant Income and Employment prediction tasks, `ACSIncome` and `ACSEmployment` (Ding et al., 2021), which are based on data from the American Community Survey's Public Use Microdata Sample (ACS PUMS). Our framework prioritizes tabular datasets and specifically leverages the ACS datasets because of their prevalence in fair FL research. According to a recent survey (Salazar et al., 2024), only 11 out of 47 fairness-aware FL methods support image data (out of these, 7 work both on image and tabular datasets), 1 out of 47 supports video data, and none support text data. With the majority (66%) of fair FL methods supporting exclusively tabular data, focusing on this data type ensures FeDa4Fair has the highest immediate research impact. Furthermore, the ACS datasets offer a unique natural horizontal partitioning (by 50 U.S. states + Puerto Rico), providing a realistic simulation of a cross-silo FL setting, which can be further partitioned to simulate realistic cross-device settings.

While our primary focus is on tabular datasets, FeDa4Fair also offers a dataset-agnostic approach to evaluating existing biases and exacerbating them to achieve a wide range of bias-heterogeneous client scenarios. This is possible because we built on top of the Flower `FederatedDataset` class[3], ensuring straightforward integration with Flower (Beutel et al., 2020), one of the most common FL frameworks used both in research and industry. Moreover, thanks to the integration of Flower with the HuggingFace datasets Hub (Lhoest et al., 2021), FeDa4Fair can also be applied with other datasets regardless of their modalities. To use this FeDa4Fair feature, practitioners provide a dataset and specify the sensitive features and the target they want to consider during the process to obtain a federated dataset for their experiments. As the capability to exacerbate and evaluate biases represents the core focus and novelty of FeDa4Fair, we detail its functionality here, while we refer readers to Appendix A for an overview of the library's general setup.

Recognizing recent concerns regarding the availability and reliability of FL research datasets, we have designed FeDa4Fair to make all parameter choices to create the dataset straightforward to disclose. To further support this goal, we have implemented a datasheet generation feature that semi-automatically provides documentation of the parameters employed to create any dataset, inheriting broader information about FeDa4Fair from a fixed template. With this, we aim for reproducibility and to ensure robust validation of fair FL research.

---

[3]https://flower.ai/docs/datasets/

## 3.1 FAIRNESS SPECIFICATIONS

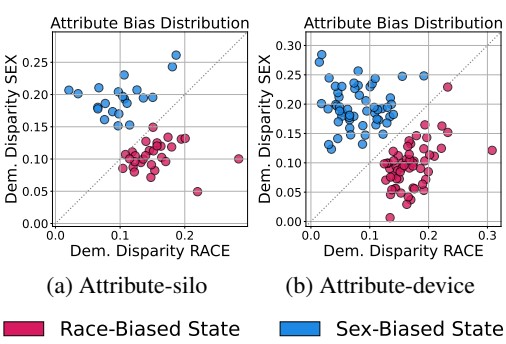

(a) Attribute-silo      (b) Attribute-device

Race-Biased State      Sex-Biased State

Figure 2: Attribute bias measured with DD on the XGBoost model for attribute benchmark datasets.

At its core, FeDa4Fair analyzes bias distribution at the client level and allows for data modification to amplify biases, facilitating exploration of different client settings while still maintaining a natural non-i.i.d. setting. A key component is the specification of the **sensitive attributes** for which bias evaluation is performed. FeDa4Fair supports both individual and intersectional fairness evaluation, allowing bias assessment based on single or combinations of attributes. For either case, the library provides fairness statistics at the client level, calculated with respect to specific attributes, attribute values, or their combinations. Currently, practitioners can choose between two fairness metrics, Demographic Disparity (DD) and Equalized Odds Difference (EOD), to perform their evaluations.

To control the granularity of fairness evaluation, users can specify a desired **fairness level**. This distinguishes between the two aforementioned client-level bias scenarios where bias varies across different `attributes` and those where it varies across different attribute `values`. At the `attribute` level, the library reports the maximum value of the selected fairness metric for each attribute. The `value` level goes beyond this by identifying both the maximum fairness metric and the specific attribute values at which this maximum occurs.

For a more detailed view, we also provide an `attribute-value` level which returns fairness metrics for all possible combinations of attributes and their values. These fairness metrics are automatically computed alongside any dataset created via FeDa4Fair, reported in tabular formats, and visualized using plots.

Figures 2 illustrate examples of *attribute-based* bias in datasets. Each dot represents a local XGBoost model trained on data from one of the 51 different states/clients of a dataset created via FeDa4Fair with `ACSIncome` as the base task (Ding et al., 2021). The plots highlight clients biased toward `RACE` (in red) and `SEX` (in blue). Similarly, Figure 3 shows the *attribute-value-based* bias in datasets showing how different clients exhibit varying degrees of

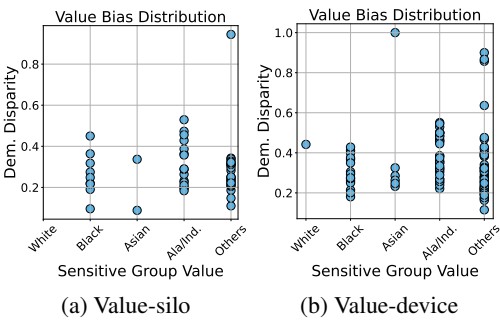

(a) Value-silo      (b) Value-device

Figure 3: Attribute value bias measured with DD on XGBoost for value benchmark datasets.

maximum DD across the values of the `RACE` attribute. Each dot represents the DD for a specific sensitive group value in a local model trained on one of the 51 clients.

In Figure 4, we present a more detailed analysis of fairness distributions for two states, now also partitioned into 5 clients each. Comparing Figures 4a and 4b, we observe distinct bias patterns across the `SEX` and `RACE` attributes. Notably, `SEX` consistently shows lower bias than `RACE` across the clients. Thus, we can easily identify the dominant bias toward specific attribute values for each client. As shown in Figure 4c, the highest level of DD spans a broad range of attribute values. For instance, in the state of "LA", the maximum DD for `RACE` occurs between $z = 4$ and $z = 8$ where $z$ denotes the sensitive attribute `RACE` and 4 and 8 are two of its possible values.

To address scenarios where the created datasets do not exhibit the desired bias distributions across clients, FeDa4Fair offers targeted **bias exacerbation** features. This feature provides researchers with fine-grained control over data manipulation, allowing them to tailor dataset properties for a wider range of fair FL research scenarios. Specifically, the bias exacerbation supports two types of manipulations: (I) label flipping and (II) datapoint dropping, both applied to data instances with

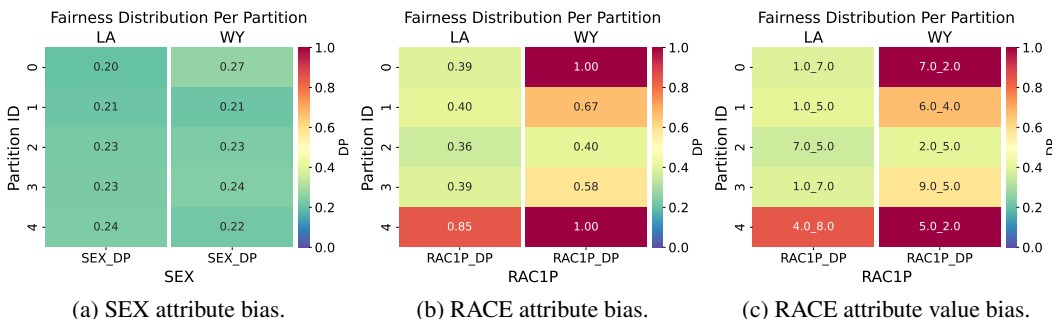

(a) SEX attribute bias.  (b) RACE attribute bias.  (c) RACE attribute value bias.

Figure 4: *Attribute* and *attribute value* bias measured with DD on the true labels and partitioning data from "LA" and "WY". These plots are generated for any dataset created with FeDa4Fair.

negative labels within selected data splits. This focus ensures these interventions amplify potential imbalances that usually impact model performance on underrepresented negative classes. To use this feature, practitioners indicate the type of manipulation to be applied to a specific combination of data splits, the sensitive attributes considered (e.g., "gender", "race"), and the attribute values to which the manipulation should apply (e.g., "female", "African American"). Furthermore, more granular control is possible by defining an extra attribute and value, e.g., only flip labels for "female" instances within the "African American" group, to enable intersectional bias manipulation. Label flipping and datapoint dropping can be applied independently or combined: for instance, flipping a percentage of negative labels for the "African American" subgroup while dropping a different percentage of datapoints with negative labels for the "Asian" subgroup.

This flexibility allows FeDa4Fair to create datasets reflecting realistic and complex bias patterns, facilitating the development and the evaluation of fair FL solutions. However, we stress that manipulated data is, of course, no longer representative of the state-level distributions and demographics in the U.S. and Puerto Rico; we strongly advise against employing data manipulated with FeDa4Fair for anything but the testing and development of federated bias mitigation strategies.

## 4 BENCHMARK DATASETS

To offer practitioners an easy access to the functionalities of FeDa4Fair, we release four benchmark datasets with realistic bias conditions constructed with FeDa4Fair: (I) an **attribute**-level biased dataset for the cross-**silo** setting (**attribute-silo**); (II) a **value**-level biased dataset for the cross-**silo** setting (**value-silo**); (III) a **attribute**-level biased dataset for the cross-**device** setting (**attribute-device**); (IV) a **value**-level biased dataset for the cross-**device** setting (**value-device**). These datasets enable a comprehensive evaluation of fair FL methods under diverse bias conditions.

All four datasets are based on the 2018 "ACSIncome" dataset (Ding et al., 2021) (see Section 3.1), and pre-partitioned to simulate the FL clients. The cross-silo datasets (**attribute-silo** and **value-silo**) use the original 51 state-wise division. For the cross-device setting (**attribute-device** and **value-device**), we increased the number of splits to 100 and 111 to reflect larger participant numbers. To quantify bias within each client of the benchmark datasets, we evaluated performance on two ML models trained for each client dataset: Logistic Regression (Cox, 1958) and XGBoost (Chen & Guestrin, 2016), in line with experimentation from (Ding et al., 2021). We measured DD as our primary fairness metric, but also reported EOD results for completeness. We consider a dataset biased toward a specific attribute or attribute value if both models exhibited the maximum DD value for the same attribute/value, and the minimum of these maximum DD values exceeded 0.09. To maintain consistency across all datasets, we iteratively increased the percentage of datapoints to be dropped through our bias exacerbation features as explained in Section 3.1. In the following, we describe each published[4] dataset's key characteristics. Additional bias-heterogeneous client settings can be subsampled from these published datasets.

---

[4]Datasets and the code they were generated with can be found at https://anonymous.4open.science/r/FeDa4Fair-AEB3/

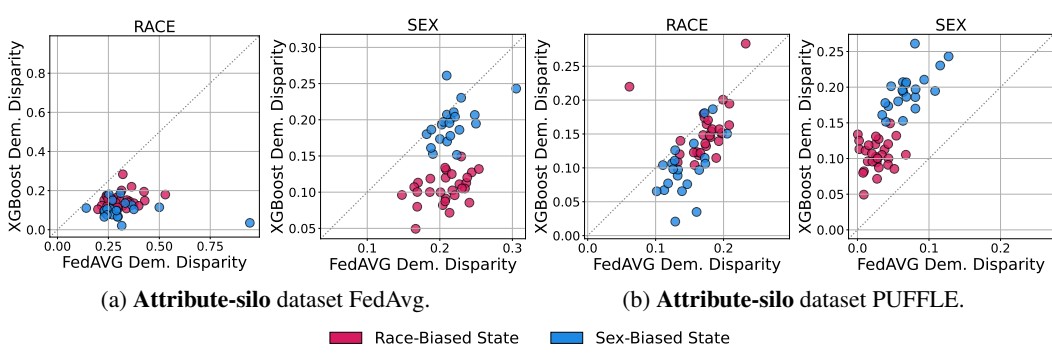

Figure 5: Attribute bias toward `RACE` and `SEX` measured with DD on the XGBoost model vs. the FedAvg model and vs. PUFFLE for the attribute-silo dataset.

**Attribute-silo dataset.** In the attribute bias-heterogeneous cross-silo setting, we leverage the natural distribution of data across the 50 states and Puerto Rico, focusing on the `RACE` and `SEX` attributes. To align with binary-focused fair FL methods (Corbucci et al., 2024; Papadaki et al., 2022; Abay et al., 2020), we binarize `RACE` into "White" and "Others". To meet the defined bias threshold, we iteratively increase the drop rate of the more biased class between `RACE` and `SEX`. Figure 2a illustrates the resulting bias distributions after these modifications. In our evaluation, 21 states exhibit a higher DD value for `SEX`, while 30 states show a higher DD value for `RACE` across both models. Further details, including the list of affected states, EOD statistics, and applied data modifications, are provided in Table 1, and Table 3 in Appendix B.

**Attribute-device dataset.** This dataset is derived from the attribute-silo dataset by splitting each state into six subsets. We then sample from these subsets to create datasets satisfying our bias constraints. As a result, the modifications applied to these datasets are directly inherited from the corresponding parent state dataset in the attribute-silo setting (Table 1). Across these subsets, we observe that 55 states exhibit stronger bias toward `SEX` and 56 toward `RACE`. These patterns are visualized in Figure 2b and summarized in Table 4.

**Value-silo dataset.** As with the Attribute-silo dataset, the value-silo is built on the natural distribution of the "ACSIncome" dataset across 50 U.S. states and Puerto Rico. However, in this case, to better reflect real-world patterns and to induce different distributions of attribute value unfairness, we avoided binarizing the sensitive attribute. This choice is motivated by the observation that bias is often present in only a subset of the attributes' values. Moreover, introducing bias into groups that are historically not affected would be inappropriate and undesirable. Instead, we considered multiple classes for the `RACE` attribute ("White", "Black", "Asian", "Alaska Native/American Indian", "Others") as a basis for analysis. Within these, we identify the attribute values exhibiting the highest DD values and apply datapoint dropping to amplify existing biases where necessary (see Table 2 in Appendix B). The resulting value-level bias distribution can be found in Figure 3a in Appendix E.2. In our analysis, we observe the most biased group is "Black" in 9 states, "Asian" in 2 states, "Alaska/Indian" in 16 states, and "Others" in 24 states (see Table 5 in Appendix 4).

**Value-device dataset.** This dataset is derived by partitioning the value-silo dataset into four subsets per state and sampling from those that satisfy our bias constraints. The resulting client-level value bias distribution includes 1 state with predominant "White" bias, 15 states with "Black" bias, 6 with "Asian" bias, 31 with "Alaska Native/American Indian" bias, and 47 with "Others" biases as shown in Figure 3b. Further details can be found in Table 2 and 6 in Appendix B.

## 5 BENCHMARK EXPERIMENTS

To contextualize our benchmark datasets, we present experiments on all four of them in the FL context by training two models: (I) a vanilla FedAvg (McMahan et al., 2017) baseline without fairness mitigation, and (II) a FedAvg model with PUFFLE (Corbucci et al., 2024) as a baseline that incorporates fairness mitigation. By comparing these scenarios, we illustrate how standard and

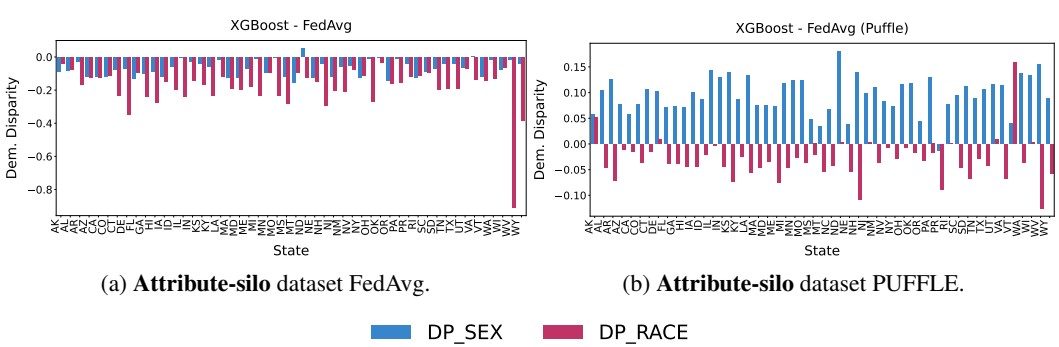

(a) **Attribute-silo** dataset FedAvg.

(b) **Attribute-silo** dataset PUFFLE.

Figure 6: Individual, per attribute, bias differences in DD between the local XGBoost models vs. the FedAvg model and vs. PUFFLE.

fairness-mitigated FL behave differently across our bias-controlled scenarios. Our aim is not to exhaustively evaluate all fair FL solutions, but to provide an example of how our benchmark dataset can be applied.

For the baseline model, we apply the FedAvg algorithm, simulating the FL training with Flower, as discussed in Section 2.1. When training the fair-mitigated FL model, we rely on PUFFLE (Corbucci et al., 2024), an in-process method to reduce model unfairness measured with DD. Specifically, during training, selected clients compute the gradient and model fairness on a given batch. The computed fairness metric is then incorporated as an additional regularization term in the model update to mitigate unfairness, by summing and weighting using a hyperparameter $\lambda$ indicating the importance of the model's utility and its fairness. Here, a $\lambda$ closer to 1 prioritizes fairness. In our experiments, we treat $\lambda$ as a hyperparameter; more details about this are reported in Appendix C. For PUFFLE, we apply unfairness mitigation for the SEX sensitive attribute with target DD=0.05. This choice allows us to evaluate two distinct scenarios. In the attribute-silo and attribute-device datasets, where SEX is one of the biased attributes, we assess how mitigation affects the disparity for both SEX-biased clients and RACE-biased clients. In contrast, in the value-silo and value-device datasets, we show how mitigating unfairness for SEX influences value bias shifts across the clients.

**Evaluation.** We assess our methodology around two principles: quantifying fairness both at the individual client level and analysing how bias distribution shifts across clients, both before and after the application of a fair FL method. In our analysis, we distinguish between cross-silo and cross-device FL scenarios. In cross-silo settings, typically, clients possess sufficient data to perform a local train/test split. Therefore, we train individual client-specific models and compare their performance to that of the global FL model. Doing so, we can assess whether participation in FL mitigates bias for specific demographic groups or individuals or conversely, if the existing bias is propagated or even exacerbated. In contrast, the cross-device setting involves clients with limited local data, which is insufficient for training robust local models. Here, FL represents the only feasible way to obtain a usable model. In these scenarios, a common practice is to partition clients into a train and a test group. However, a challenge arises: local fairness metrics can only be computed on true labels or external model predictions. A potential solution is to compare the fairness metrics on test clients using the FL model against those obtained from a set of external models trained on similar datasets, possibly provided by the FL orchestrator. However, we leave this for future discussion and choose to evaluate the cross-device settings using the same approach as the cross-silo settings. Specifically, we evaluate fairness on the test clients, using the model trained during the FL process.

**Attribute bias benchmarks.** The **group-level** results for the **attribute-silo** dataset are illustrated in Figure 5. Each dot represents a client, showing its DD value with the local XGBoost model (Y axis) and FedAvg model or PUFFLE model (X axis). Points below the diagonal line indicate an increase in DD, i.e., an increase in unfairness after FL training. On the contrary, dots above the diagonal suggest greater unfairness with the local XGBoost model. As shown in Figure 5a, training with FedAvg leads to an increase in DD for both RACE and SEX across most clients, highlighting the issue of bias propagation (Chang & Shokri, 2023) in FL. In contrast, Figure 5b shows how PUFFLE effectively enforces fairness constraints on the SEX attribute, resulting in a SEX disparity reduction for all clients.

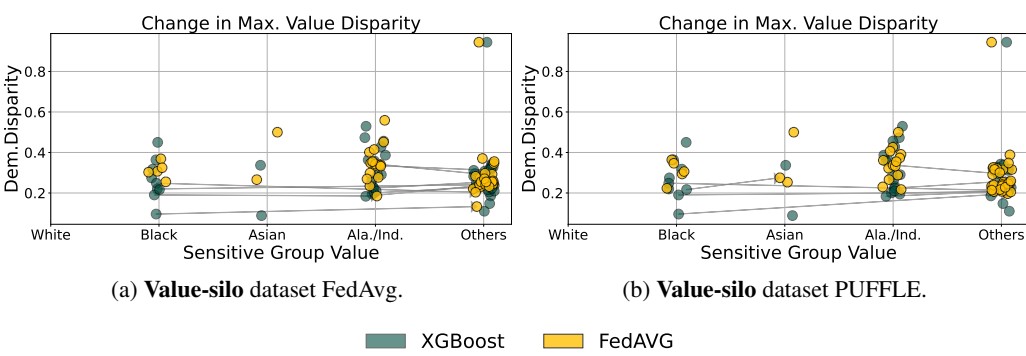

Figure 7: Attribute value bias toward `RACE` as well as value changes measured with DD on the XGBoost model vs. the FedAvg model and vs. PUFFLE for the value-silo dataset.

However, this comes at the cost of increased `RACE` disparity compared to the local XGBoost model. Thus, states biased towards the `RACE` attribute profit less, in terms of bias reduction on their most biased attribute, from participating in PUFFLE. Figure 6a shows a **client-level** evaluation comparing the XGBoost local model with the FedAvg model. Here, FedAvg always increases the unfairness of the individual clients. In contrast, when using PUFFLE (Figure 6b), all clients benefit in terms of `SEX` unfairness mitigation and some, such as "AK", "VT", also in terms of `RACE` unfairness mitigation. This trend remains consistent when using Logistic Regression as the local model (Figure 10, 16 in Appendix E.1). Additional results for the **attribute-device** dataset are detailed in Appendix E.1.

**Value bias benchmarks.** For the **value-silo** dataset, Figure 7 shows not only how DD changes when training the model with FL but also how the distribution of the attribute value associated with the maximum DD shifts. We observe a clear trend: underrepresented groups (in this case, "Alaska Native/American Indian" and "Others") tend to be disproportionately harmed from FL participation. Clusters, corresponding to "Others", grow in density when training the FL model, indicating how this group becomes more often associated with the highest DD. Under PUFFLE, this effect becomes more pronounced: with clients increasingly reporting "Others" as the highest DD value. The overall `RACE` disparity only slightly improves on a global visualization (for the individual level see Figure 19 in Appendix E.2). The same patterns hold when using Logistic Regression as local models (Figure 15, 20 in Appendix E.2).

Overall, we observe consistent results across both cross-silo and cross-device experiments for each type of bias-heterogeneous client setting. However, the specific bias distribution, i.e., attribute or attribute value, led to substantially different outcomes. This highlights the importance of evaluating fair FL solutions across diverse scenarios to ensure robust and fair performance.

## 6 CONCLUSION

Current fairness evaluation in FL is usually based on the assumption of a uniform bias distribution across the clients. This often creates an illusion of fairness at the global level while ignoring the complex, heterogeneous biases that exist at the individual client's level. Current state-of-the-art solutions and evaluations are insufficient in realistic scenarios involving value and attribute bias.

To address this, we introduced FeDa4Fair, the first library designed to create datasets for FL experimentation with heterogeneous client bias. By providing a reproducible and extensible framework, FeDa4Fair is a crucial first step towards enabling more robust and systematic evaluation of fair FL methodologies.

Thanks to its extensibility, FeDa4Fair is designed to support the continual expansion and diversification of the fairness evaluation landscape in FL. This facilitates two critical directions for future research: first, the inclusion of even more complex real-world scenarios and second, the investigation of fairness in cross-device settings where clients possess only limited data.

ETHICS STATEMENT

FeDa4Fair allows for manipulating datasets such that they reflect more complex bias patterns to facilitate fair FL method evaluation and to improve bias mitigation development. However, we stress that manipulated data is, of course, not anymore representative of the state-level distributions and demographics in the U.S. and Puerto Rico. We strongly advise against employing data manipulated with FeDa4Fair for anything but the testing and development of federated bias mitigation strategies.

REPRODUCIBILITY

We provide our implementation at `https://anonymous.4open.science/r/FeDa4Fair-AEB3/`. The benchmark datasets can be found in the data file provided there including the code for their generation `https://anonymous.4open.science/r/FeDa4Fair-AEB3/src/FeDa4Fair/creating_datasets.py`. A well-documented example is also provided at `https://anonymous.4open.science/r/FeDa4Fair-AEB3/src/FeDa4Fair/example.ipynb`. Hyperparameters are detailed in Section C. Furthermore, generation of the datasets and model training should be possible on most commercial laptops.

LLM USAGE

LLMs were used to aid non-native speakers with grammar and word corrections.

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

## A  FeDa4Fair general setup

We rely on several parameters as a general setup, which are independent of fairness specifications. Specifically, this implies that FeDa4Fair can also create data for analyzing standard FL methods without focus on fairness. Therefore, FeDa4Fair can provide datasets for a wide range of FL methods.

**Underlying dataset.** For dataset customization, users can specify the underlying dataset to load, currently supporting `ACSIncome` and `ACSEmployment` (Ding et al., 2021). Both datasets are extracted from the Public Use Microdata Sample of the American Community Survey (ACS) across all 50 states of the USA and Puerto Rico between 2014 and 2018. For `ACSIncome`, the task involves predicting if an individual's income is above $50,000$ or not. In `ACSEmployment`, the task involves predicting an individual's employment status. While loading these datasets, it is possible to specify the year of the collected data (2014-2018) and the time horizon of the ACS sample ("`1-Year`" or "`5-Year`"). Furthermore, users may define a list of U.S. states to include; if none are specified, all available states will be loaded. Here, specifically, we keep the natural division of the dataset into 51 entities to exploit the natural non-i.i.d. distribution of the samples.

**Preprocessing.** For preprocessing, users may optionally provide a remapping dictionary to modify categorical features or labels. Also, this provides the possibility to create datasets with binarized sensitive attributes. The latter is of importance, as many fair FL methods only consider binary sensitive attributes (Salazar et al., 2024).

**FL setting.** As introduced in Section 2.1, FL scenarios can be categorized into cross-silo and cross-device settings. To account for the different client numbers and train/test settings, we implemented the following. Firstly, the framework allows users to partition data from each state into multiple subsets in a custom way. For this, a user-defined `Partitioner` object defines the partitioning strategy and number of partitions for each state. This stems from the Flower framework (Beutel et al., 2020), and a vast range of partitioning strategies are supported here (e.g., Dirichlet Partitioner, Linear Partitioner). If, instead of a `Partitioner` object, only the number of partitions is provided, this defaults to splitting the state data into IID partitions. Secondly, the framework supports different strategies to split the data into training and testing subsets. In the cross-silo setting, each client receives its partition of the dataset, which is further divided into training, validation, and test sets. Additionally, users can define the proportion of data allocated to each subset in the cross-silo scenario. For the cross-device setting, we recommend generating the dataset first and then partitioning the clients into separate groups for training and testing. If no splitting strategy is specified, the entire dataset remains intact, and train/test splitting can be managed externally.

## B  Benchmarking datasets

In the scope of this paper, we publish four datasets covering different bias scenarios to explore the behavior of fair FL methods:

   (I)  an **attribute-silo** dataset: attribute-level biased dataset for the cross-silo setting;

  (II)  a **value-silo** dataset: value-level biased dataset for the cross-silo setting;

 (III)  an **attribute-device** dataset: attribute-level biased dataset for the cross-device setting;

  (IV)  a **value-device** dataset: value-level biased dataset for the cross-device setting.

We provide the concrete modifications applied for bias exacerbation in Table 1 for datasets (I) and (III) and in Table 2 for datasets (II) and (IV). Furthermore, for each of these four datasets, we listed the counts and states sorted by where they take on the maximal bias in Table 3, 5, 4, and 6 respectively. Additionally, in Figures 8 and 9, we report the Demographic Disparity distribution for each of the four datasets on Logistic Regression models.

## C  Hyperparameter Tuning

To perform hyperparameter tuning when training the FedAvg baseline, we performed a Bayesian optimization to maximize the model validation accuracy. The parameters that we optimised are: Learning Rate, Batch Size, Optimiser, and number of local epochs. When training the FL model with

PUFFLE[5], we followed the suggestions reported in the paper by Corbucci et al. (2024). Therefore, we performed a Bayesian optimisation to minimise the model validation accuracy while keeping the model unfairness under $T = 0.05$. In this case, we optimised Learning Rate, Batch Size, optimiser, number of local epochs, and the value of the $\lambda$ used for the unfairness mitigation.

# D  HARDWARE

For all model-fitting experiments presented in this paper, we used a NVIDIA DGX H100 with 224 CPU Intel(R) Xeon(R) Platinum 8480CL, 8 GPUs Nvidia H100 (80GB) and 2TB of RAM. We employed this machine for convenience while fitting models: a single CPU will suffice, no GPU is necessary, and the memory requirements are in line with what is available on most commercial laptops. We here include the machine's characteristics for reproducibility purposes, but we believe that most resource-constrained laboratories will be able to create data with FeDa4Fair. Model fitting on our tabular datasets is similarly cheap. Moreover, to avoid the need to create the dataset, we also provide 4 datasets to download. In the direction of reducing the computational needs and testing the fairness-aware FL in resource-constrained scenarios, it is possible to reduce the size of these fixed datasets by only subsampling a set of client datasets. We provide extensive statistics on each client dataset so that users may make well-founded decisions on which datasets to sample. Additionally, if restrictions on computation exist and practitioners want to create their dataset, it is not necessary to download the full ACS datasets, but only, e.g., preselected states.

# E  ADDITIONAL EXPERIMENTS

In line with Section 5, we provide additional experiments here on the cross-device datasets as well as on Logistic Regression as an additional local model.

## E.1  ADDITIONAL ATTRIBUTE BIAS BENCHMARK EXPERIMENTS

Group-level results for the attribute-device dataset are shown in Figures 11 and 12. Each point represents a dataset, plotting its Demographic Disparity (DD) value with local and FL models with either the global FedAvg model or PUFFLE. Points below the diagonal indicate increased DD after training (higher unfairness), while those above show higher unfairness before training. For PUFFLE, unfairness mitigation was applied specifically to the SEX attribute.

Figures 11a and 12a demonstrate that FedAvg training often propagates bias, with DD increasing for both RACE and SEX across most datasets. In contrast, Figures 11b and 12b reveal that PUFFLE's fairness constraints on SEX lead to reductions in SEX disparity across all clients. While some clients experience an increase in RACE disparity with PUFFLE, it's unclear whether there exists a specific client group that benefits from improvements across both attributes. PUFFLE's performance on the RACE attribute is comparable to FedAvg training without fairness constraints.

For client-level evaluation, Figures 17 and 18 show that PUFFLE improves fairness on both SEX and RACE for some clients, and on SEX alone for others, highlighting the impact of the chosen sensitive attribute within the PUFFLE pipeline.

## E.2  ADDITIONAL VALUE BIAS BENCHMARK EXPERIMENTS

For the value-device dataset, Figures 13 and 14 show results obtained when comparing FedAvg and PUFFLE models, respectively, with XGBoost and Logistic regression local models. In particular, the plots highlight changes in Demographic Disparity (DD), and also how the distribution of values with maximal DD shifts. The trend indicates that underrepresented groups (here, values "Alaska Native/American Indian" and "Others") experience a greater disadvantage from federated training. Their clusters grow, particularly for value "Others", while others shrink.

However, overall RACE disparity appears to improve slightly in the group visualization. Further analysis at the individual level in Figures 21 and 22 underline this, as we can see a clear indication that RACE disparity slightly improves for some clients and stays stable for others.

---

[5]PUFFLE GitHub repository: `https://github.com/lucacorbucci/PUFFLE`

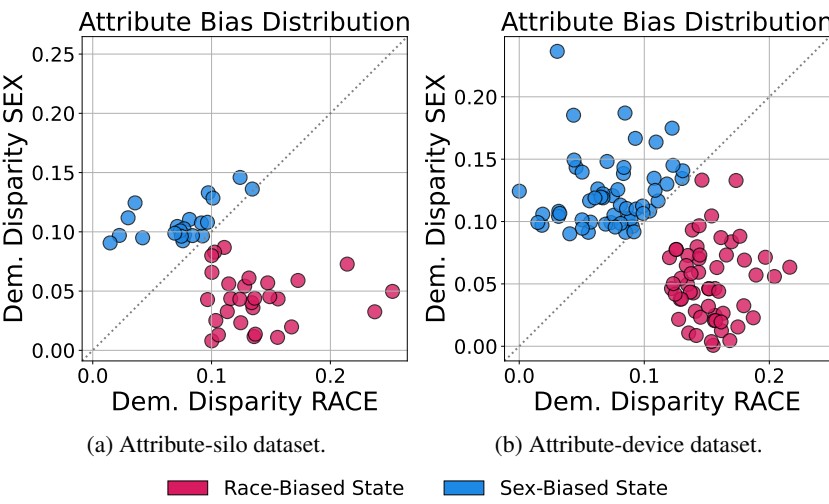

(a) Attribute-silo dataset.  (b) Attribute-device dataset.

Figure 8: Attribute bias measured with Demographic Disparity on the Logistic Regression model for the four benchmark datasets.

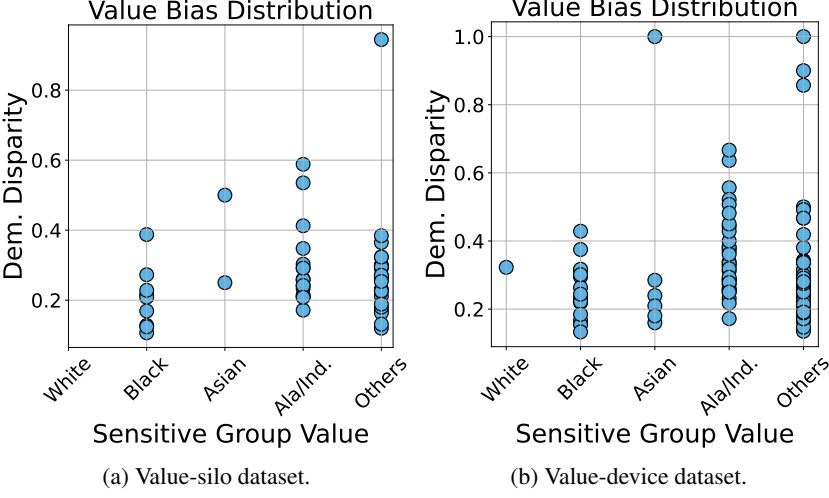

(a) Value-silo dataset.  (b) Value-device dataset.

Figure 9: Attribute value bias measured with Demographic Disparity on the Logistic Regression model for the four benchmark datasets.

Table 1: Attribute-silo, attribute-device dataset: Applied Modifications.

| State | Drop Rate | Attribute | Value |
|-------|-----------|-----------|-------|
| WY | 0.1 | SEX | 2 |
| WI | 0.1 | RAC1P | 2 |
| ND | 0.1 | SEX | 2 |
| CA | 0.1 | RAC1P | 2 |
| MT | 0.1 | RAC1P | 2 |
| LA | 0.1 | SEX | 2 |
| KY | 0.1 | RAC1P | 2 |
| ME | 0.1 | RAC1P | 2 |
| AL | 0.2 | RAC1P | 2 |
| IN | 0.2 | SEX | 2 |
| MS | 0.2 | RAC1P | 2 |
| GA | 0.2 | RAC1P | 2 |
| VT | 0.2 | RAC1P | 2 |
| IL | 0.3 | SEX | 2 |
| WA | 0.3 | SEX | 2 |
| NH | 0.3 | SEX | 2 |
| PA | 0.4 | SEX | 2 |
| WV | 0.4 | SEX | 2 |
| AR | 0.4 | SEX | 2 |
| KS | 0.4 | SEX | 2 |
| OR | 0.4 | RAC1P | 2 |
| TX | 0.4 | SEX | 2 |
| DE | 0.4 | RAC1P | 2 |
| OK | 0.4 | SEX | 2 |
| ID | 0.4 | SEX | 2 |
| MI | 0.5 | SEX | 2 |
| VA | 0.5 | SEX | 2 |
| TN | 0.5 | SEX | 2 |
| OH | 0.5 | SEX | 2 |
| MO | 0.6 | SEX | 2 |
| PR | 0.6 | SEX | 2 |

Table 2: Value-silo, value-device dataset: Applied Modifications.

| State | Drop Rate | Attribute | Value |
|-------|-----------|-----------|-------|
| AZ | 0.1 | RAC1P | 5 |
| OH | 0.1 | RAC1P | 4 |
| AR | 0.2 | RAC1P | 4 |
| MN | 0.2 | RAC1P | 5 |
| OR | 0.2 | RAC1P | 2 |
| WV | 0.2 | RAC1P | 5 |
| DE | 0.3 | RAC1P | 4 |
| LA | 0.3 | RAC1P | 5 |
| NE | 0.3 | RAC1P | 4 |
| AK | 0.5 | RAC1P | 4 |
| MS | 0.5 | RAC1P | 4 |
| PR | 0.6 | RAC1P | 4 |

Table 3: **Attribute-silo dataset**: Counts and states for which the maximum Demographic Disparity/Equalized Odds Difference across all evaluated models is reached for the stated sensitive attribute. States where bias is distributed the same for both Demographic Disparity/Equalized Odds Difference are marked in bold.

| Sensitive att. | Demographic Disparity | | Equalized Odds Difference | |
|---|---|---|---|---|
| | Count | States | Count | States |
| SEX | 21 | AR, ID, IL, IN, KS, LA, MI, MO, ND, NH, OH, OK, PA, SD, TN, TX, UT, VA, WA, WV, WY | 0 | |
| RACE | 29 | **AK**, **AL**, **AZ**, CA, CO, CT, DE, FL, GA, **HI**, IA, KY, MA, MD, ME, **MN**, **MS**, MT, NC, **NE**, NJ, **NM**, **NV**, **NY**, OR, **RI**, SC, **VT**, **WI** | 17 | **AK**, **AL**, **AZ**, **HI**, **MN**, **MS**, ND, **NE**, **NM**, **NV**, **NY**, OK, **RI**, SC, **VT**, **WI**, WY |

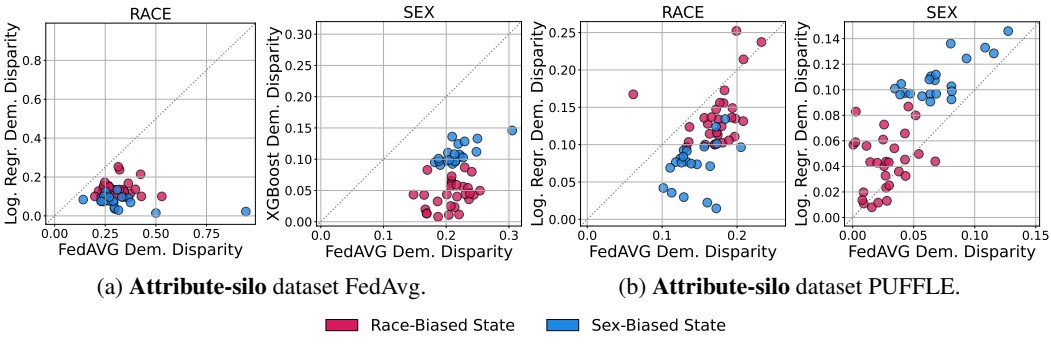

(a) **Attribute-silo** dataset FedAvg.  (b) **Attribute-silo** dataset PUFFLE.

■ Race-Biased State    ■ Sex-Biased State

Figure 10: Attribute bias toward RACE and SEX measured with Demographic Disparity on the Logistic Regression model vs. the FedAvg model and vs. PUFFLE for the attribute-silo dataset.

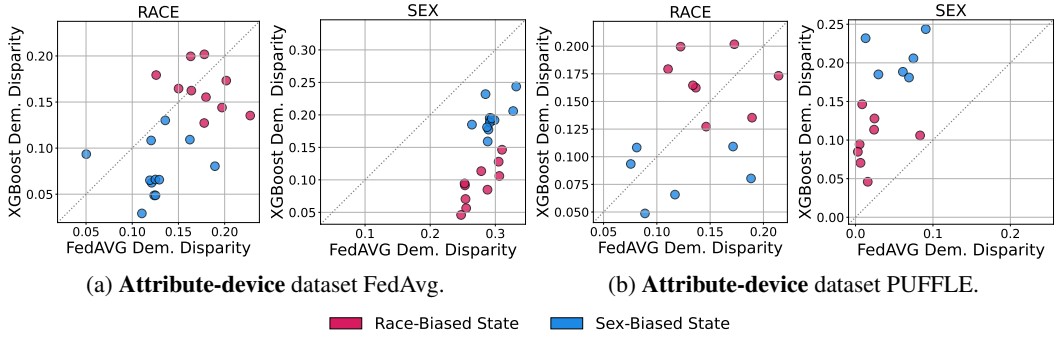

(a) **Attribute-device** dataset FedAvg.  (b) **Attribute-device** dataset PUFFLE.

■ Race-Biased State    ■ Sex-Biased State

Figure 11: Attribute bias toward RACE and SEX measured with Demographic Disparity on the XGBoost model vs. the FedAvg model and vs. the PUFFLE model for the attribute-silo dataset.

Table 4: **Attribute-device dataset**: Counts and states for which the maximum Demographic Disparity/Equalized Odds Difference across all evaluated models is reached for the stated sensitive attribute. States where bias is distributed the same for both Demographic Disparity/Equalized Odds are marked in bold.

| Sensitive att. | Demographic Disparity | | Equalized Odds Difference | |
|---|---|---|---|---|
| | Count | States | Count | States |
| SEX | 55 | ID_0, IL_2, IL_3, IL_4, IN_0, IN_2, IN_3, **KS_2**, KS_4, LA_4, MI_0, MI_1, MI_2, MI_3, MI_4, MO_1, MO_4, ND_1, **ND_4**, NE_5, NH_0, NH_1, **NH_4**, NH_5, OH_0, OH_1, OH_2, OH_3, OH_4, OK_0, OK_2, OK_3, PA_0, PA_1, PA_2, PA_3, PA_5, TN_2, TN_4, TN_5, TX_0, TX_1, TX_3, TX_4, TX_5, UT_0, UT_4, UT_5, VA_0, VA_1, VA_3, VA_4, WA_3, WA_5, WV_0 | 3 | **KS_2**, **ND_4**, **NH_4**, |
| RACE | 56 | **AL_2**, AL_3, AL_4, **AZ_0**, **AZ_5**, CA_1, **CO_0**, **CO_3**, **CT_2**, CT_5, FL_5, **HI_0**, **IA_0**, **IA_1**, **ID_2**, LA_2, MA_0, **MA_4**, MA_3, **ME_1**, **MN_0**, **MN_3**, **MN_4**, MN_5, **MS_4**, **MS_5**, NC_1, **NE_3**, **NE_4**, NJ_1, NJ_3, NJ_4, **NM_1**, **NM_5**, NV_0, **NV_1**, NV_3, NY_0, **NY_1**, NY_2, **NY_3**, NY_4, **NY_5**, OR_1, **OR_3**, **OR_4**, **RI_2**, **SC_0**, SC_1, SC_2, SC_4, **SD_3**, **UT_1**, WI_2, **WI_3**, **WV_5** | 39 | **AL_2**, **AZ_0**, **AZ_5**, **CO_0**, **CO_3**, **CT_2**, **HI_0**, **IA_0**, **IA_1**, **ID_2**, **MA_4**, **ME_1**, MI_2, **MN_0**, **MN_3**, **MN_4**, **MS_4**, **MS_5**, **NE_3**, **NE_4**, NH_0, NH_5, **NM_1**, **NM_5**, **NV_1**, **NY_1**, **NY_3**, **NY_5**, OK_3, **OR_3**, **OR_4**, **PA_3**, **RI_2**, **SC_0**, **SD_3**, TN_2, UT_5, **WI_3**, **WV_5** |

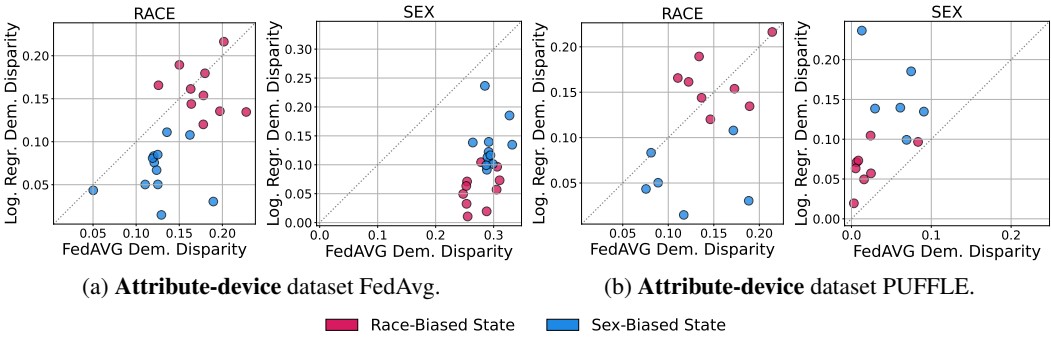

(a) **Attribute-device** dataset FedAvg.  (b) **Attribute-device** dataset PUFFLE.

merges Race-Biased State   ▮ Sex-Biased State

Figure 12: Attribute bias toward RACE and SEX measured with Demographic Disparity on the Logistic Regression model vs. the FedAvg model and vs. the PUFFLE model for the attribute-silo dataset.

Table 5: **Value-silo dataset**: Counts and states for which the maximum Demographic Disparity/Equalized Odds Difference across all evaluated models is reached for RACE and the stated value. States where bias is distributed the same for both Demographic Disparity/Equalized Odds Difference are marked in bold.

| Value (RACE) | Demographic Disparity | | Equalized Odds Difference | |
|---|---|---|---|---|
| | Count | States | Count | States |
| 1 | 0 | | 0 | |
| 2 | 9 | FL, **ME**, MN, **MT**, ND, **NH**, OK, **SD**, **VT** | 5 | **ME**, **MT**, **NH**, **SD**, **VT** |
| 3 | 2 | PR, **WY** | 2 | ND, **WY** |
| 4 | 16 | AK, **AL**, **CO**, **CT**, **DE**, **GA**, **HI**, ID, **KY**, **MS**, **NE**, **NM**, **OH**, **OR**, **PA**, UT | 17 | **AL**, **CO**, **CT**, **DE**, FL, **GA**, **HI**, **KY**, **MS**, **NE**, NJ, **NM**, NY, **OH**, **OR**, **PA**, SC |
| 5 | 24 | **AR**, AZ, **CA**, IA, **IL**, **IN**, KS, **LA**, MA, MD, **MI**, MO, **NC**, NJ, **NV**, NY, **RI**, SC, TN, **TX**, VA, WA, **WI**, **WV** | 13 | **AR**, **CA**, ID, **IL**, **IN**, **LA**, **MI**, **NC**, **NV**, **RI**, **TX**, **WI**, **WV** |

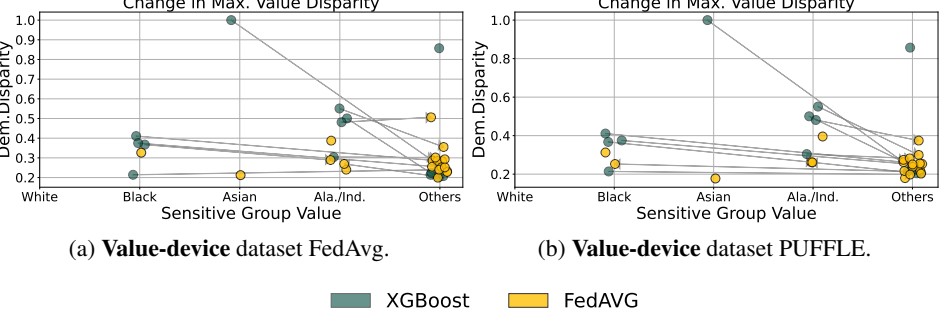

(a) **Value-device** dataset FedAvg.   (b) **Value-device** dataset PUFFLE.

XGBoost     FedAVG

Figure 13: Attribute value bias toward RACE as well as value changes measured with Demographic Disparity on the XGBoost model vs. the FedAvg model and vs. PUFFLE for the value-device datasets.

Table 6: **Value-device dataset**: Counts and states for which the maximum Demographic Disparity/Equalized Odds Difference across all evaluated models is reached for race and the stated value. States where bias is distributed the same for both Demographic Disparity/Equalized Odds Difference are marked in bold.

| Value (`RACE`) | Demographic Disparity | | Equalized Odds Difference | |
|---|---|---|---|---|
| | Count | States | Count | States |
| 1 | 1 | WV_2 | 0 | |
| 2 | 15 | **ID_1**, **ME_2**, MI_1, MN_0, MN_1, **MN_2**, **MT_2**, **ND_0**, **ND_1**, **NH_2** **SD_1**, **SD_2**, **VT_0**, WI_0, **WY_0** | 17 | HI_2, ID_0, **ID_1**, KY_0, ME_1, **ME_2**, **MN_2**, **MT_2**, **ND_0**, **ND_1**, NE_0, **NH_2**, PA_0, **SD_1**, **SD_2**, **VT_0**, **WY_0** |
| 3 | 6 | AK_0, **AR_2**, **MT_0**, MT_1, NE_1, **VT_1** | 4 | **AR_2**, **MT_0**, NE_2, **VT_1** |
| 4 | 31 | **AK_1**, AK_2, **CO_1**, **CT_1**, **CT_2**, **DE_0**, **DE_1**, **GA_1**, **GA_2**, **HI_0**, **HI_1**, **ID_2**, **IL_0**, **IL_1**, **KS_0**, **KY_2**, LA_1, MA_0, ME_1, **MI_2**, **MO_0**, **MO_2**, **MS_0**, **MS_1**, NE_0, NJ_2, NM_0, **NY_0**, **RI_0**, **SC_2**, **TN_2** | 35 | **AK_1**, AZ_2, **CO_1**, CO_2, **CT_1**, **CT_2**, **DE_0**, **DE_1**, GA_0, **GA_1**, **GA_2**, **HI_0**, **HI_1**, **ID_2**, **IL_0**, **IL_1**, IN_2, **KS_0**, **KY_2**, **MI_2**, **MO_0**, **MO_2**, **MS_0**, **MS_1**, NC_1, **NY_0**, NY_1, OH_0, **RI_0**, **SC_2**, TN_0, **TN_2**, VA_2, WI_0, WV_0 |
| 5 | 47 | AL_0, AL_2, AR_0, **AR_1**, AZ_0, **AZ_1**, AZ_2, **CA_1**, **CA_2**, CO_2, **DE_2**, FL_0, GA_0, HI_2, ID_0, **IN_0**, IN_2, KS_2, KY_0, **MD_0**, **MD_1**, **MD_2**, MO_1, NC_0, NC_1, NC_2, NE_2, NJ_1, NV_0, **NV_2**, NY_1, NY_2, OH_0, PA_0, PA_2, RI_2, **SC_1**, TN_0, TX_0, **TX_1**, **TX_2**, VA_2, WA_1, WA_2, **WI_1**, **WI_2**, WV_0 | 17 | **AR_1**, **AZ_1**, **CA_1**, **CA_2**, **DE_2**, **IN_0**, MA_0, **MD_0**, **MD_1**, **MD_2**, **NV_2**, **SC_1**, **TX_1**, **TX_2**, **WI_1**, **WI_2**, WV_2 |

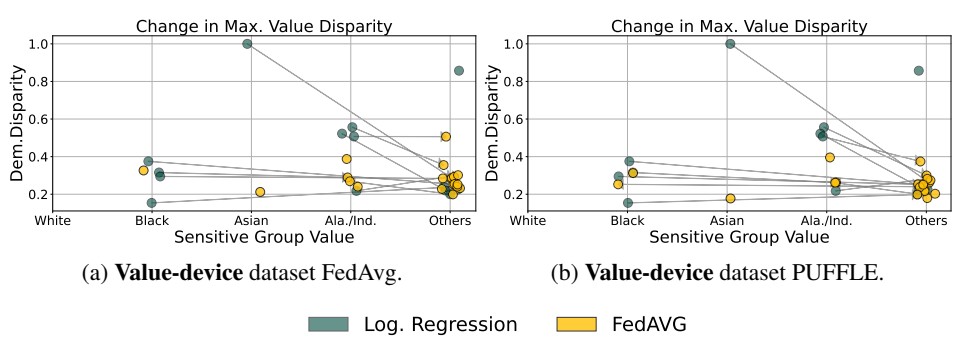

(a) **Value-device** dataset FedAvg.  (b) **Value-device** dataset PUFFLE.

Figure 14: Attribute value bias toward RACE as well as value changes measured with Demographic Disparity on the Logistic Regression model vs. the FedAvg model and vs. PUFFLE for the value-device datasets.

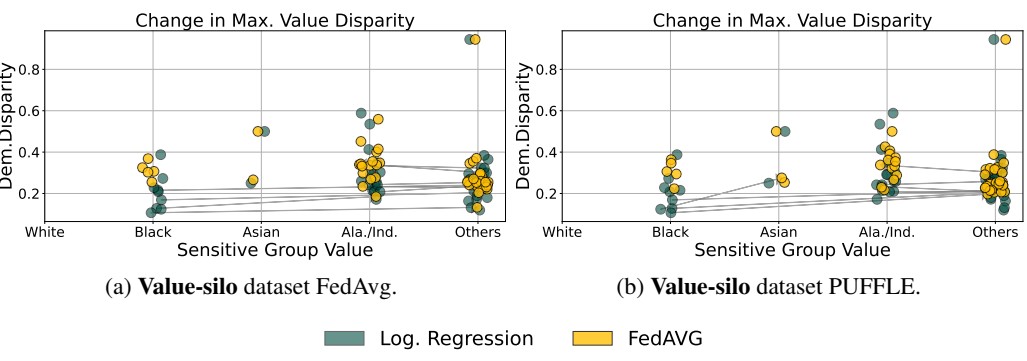

(a) **Value-silo** dataset FedAvg.  (b) **Value-silo** dataset PUFFLE.

Figure 15: Attribute value bias toward RACE as well as value changes measured with Demographic Disparity on the Logistic Regression model vs. the FedAvg model and vs. PUFFLE for the value-silo dataset.

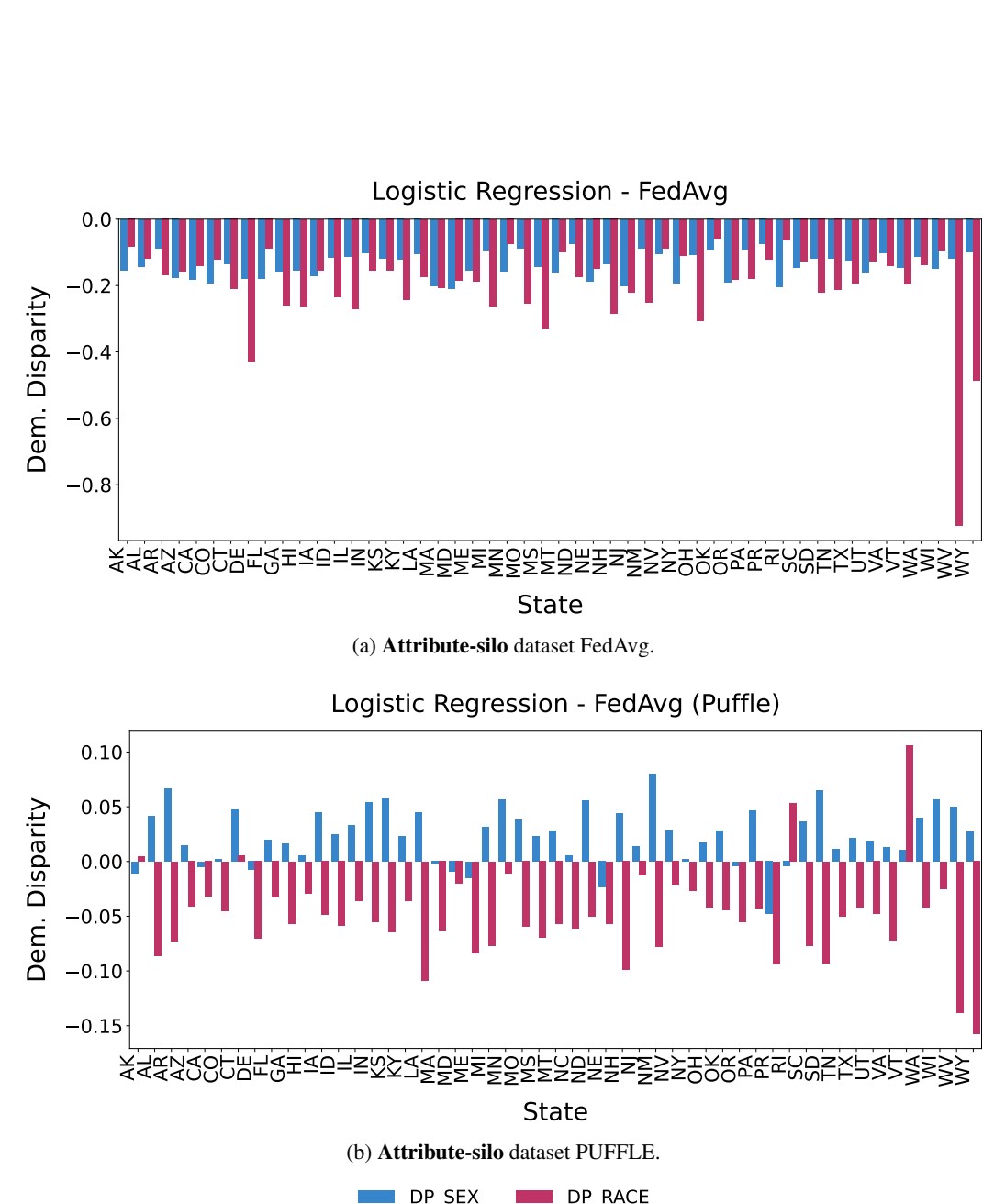

(a) **Attribute-silo** dataset FedAvg.

(b) **Attribute-silo** dataset PUFFLE.

Figure 16: Individual, per attribute, bias differences in Demographic Disparity between the local Logistic Regression models vs. the FedAvg model and vs. PUFFLE.

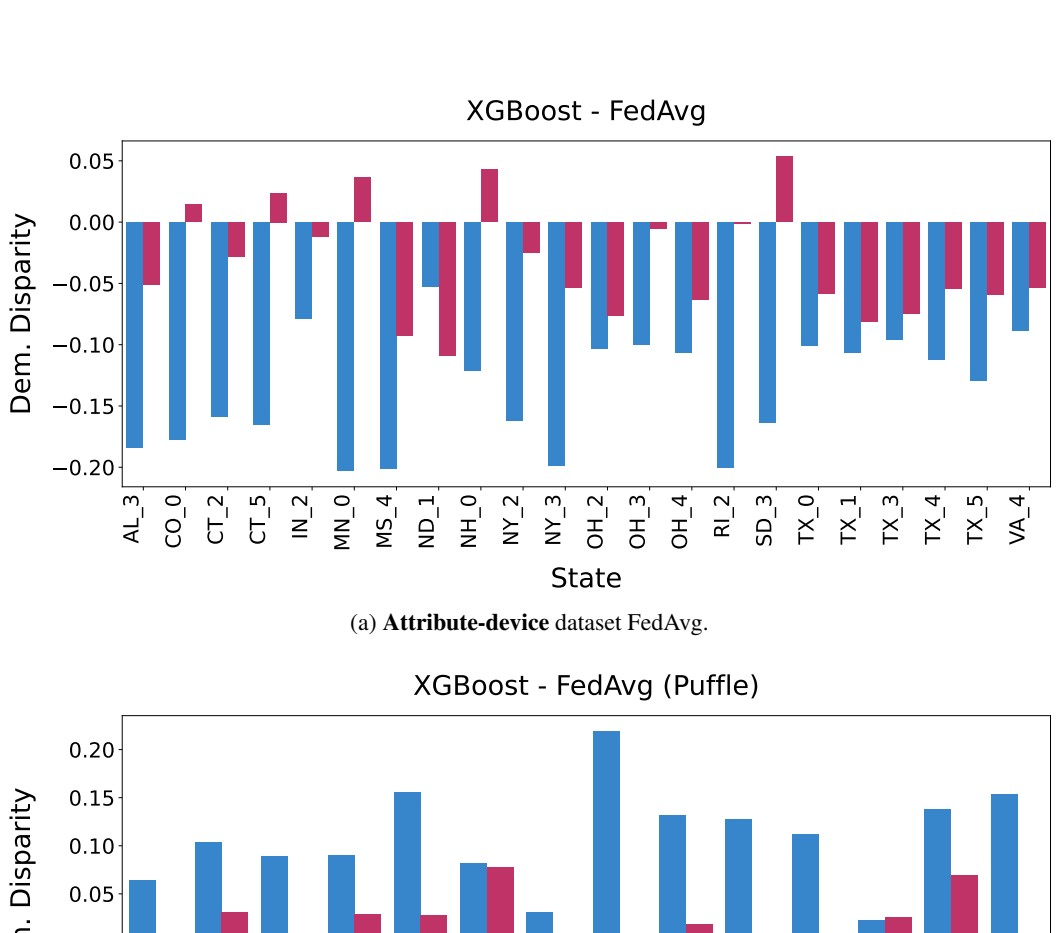

(a) **Attribute-device** dataset FedAvg.

(b) **Attribute-device** dataset PUFFLE.

Figure 17: Individual differences in Demographic Disparity between the local XGBoost models vs. the FedAvg model and vs. the PUFFLE model for the attribute-device dataset.

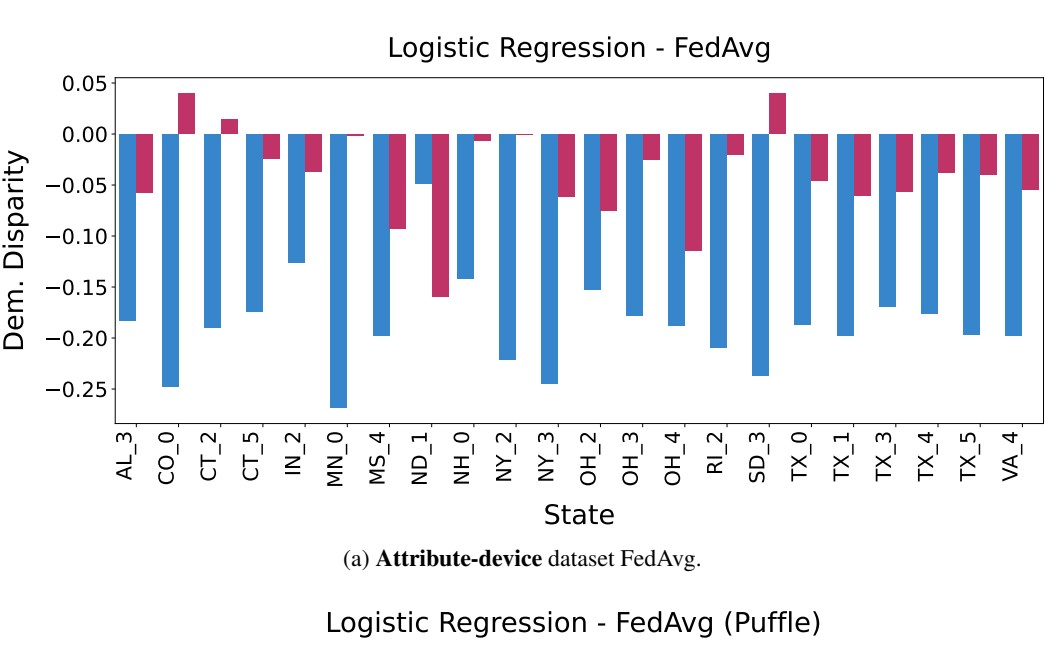

(a) **Attribute-device** dataset FedAvg.

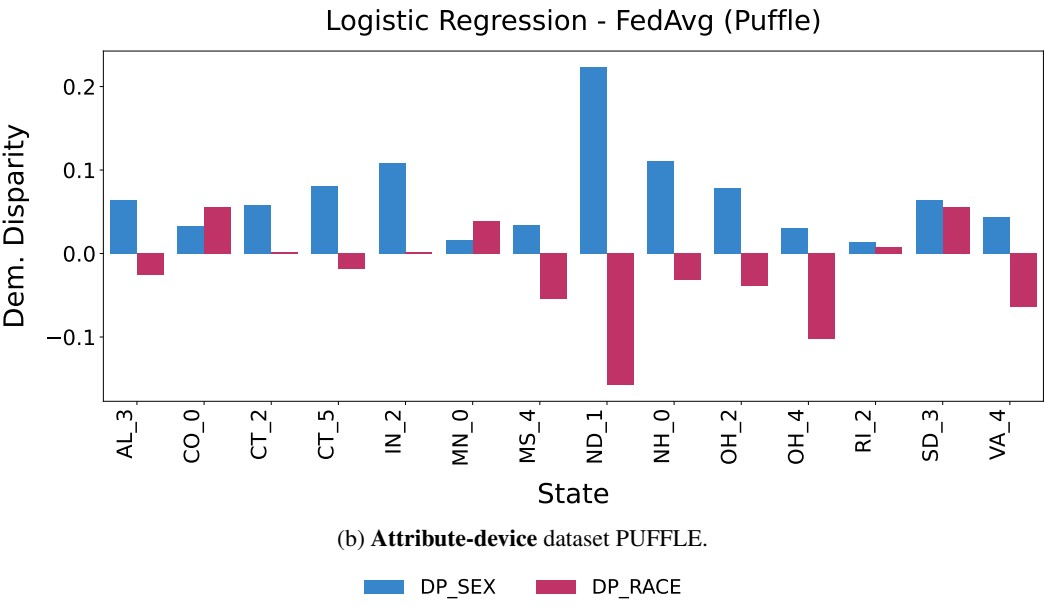

(b) **Attribute-device** dataset PUFFLE.

Figure 18: Individual differences in Demographic Disparity between the local Logistic Regression models vs. the FedAvg model and vs. the PUFFLE model for the attribute-device dataset.

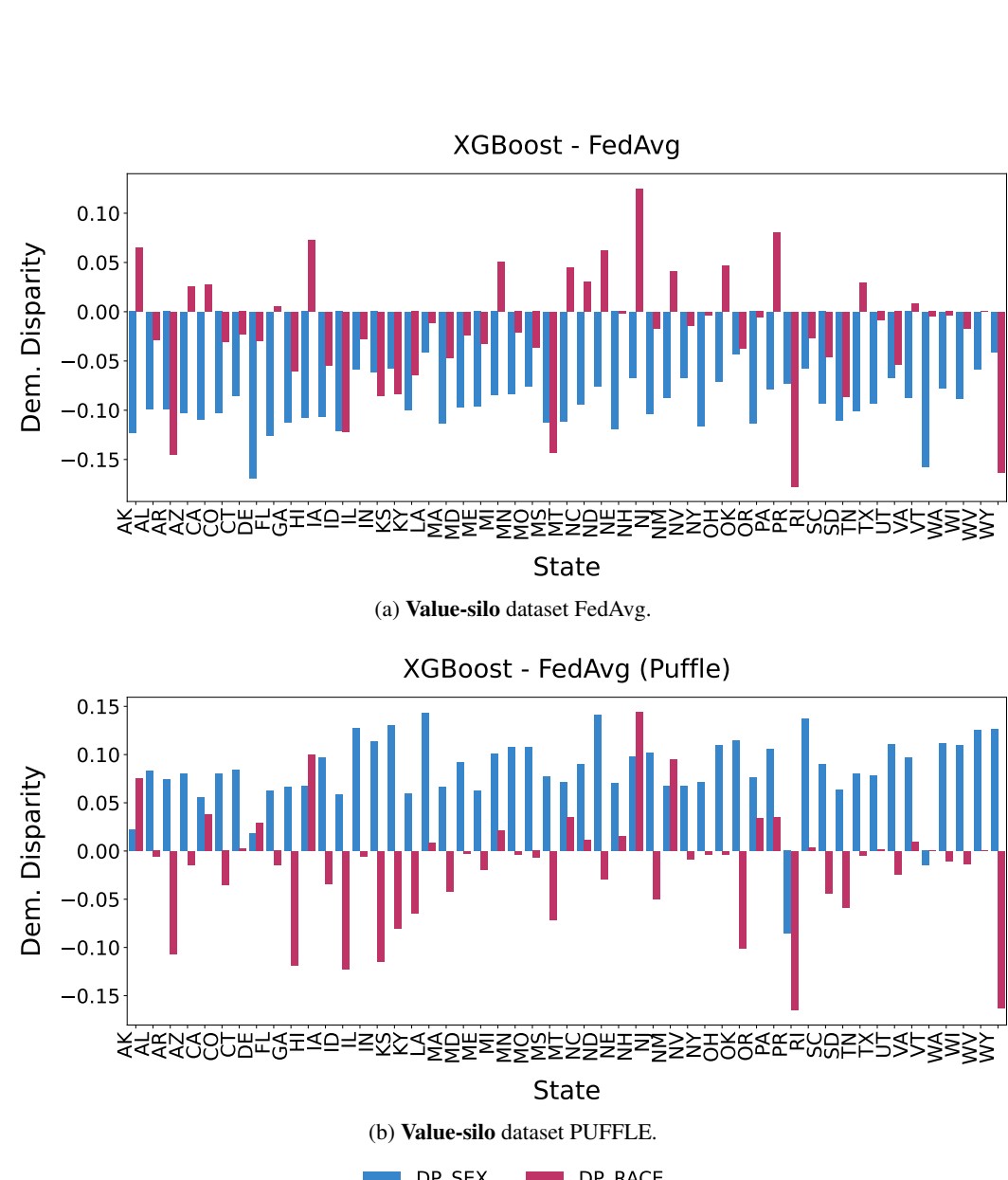

(a) **Value-silo** dataset FedAvg.

(b) **Value-silo** dataset PUFFLE.

Figure 19: Individual, per attribute, bias differences in Demographic Disparity between the local XGBoost models vs. the FedAvg model and vs. PUFFLE for the value-silo dataset.

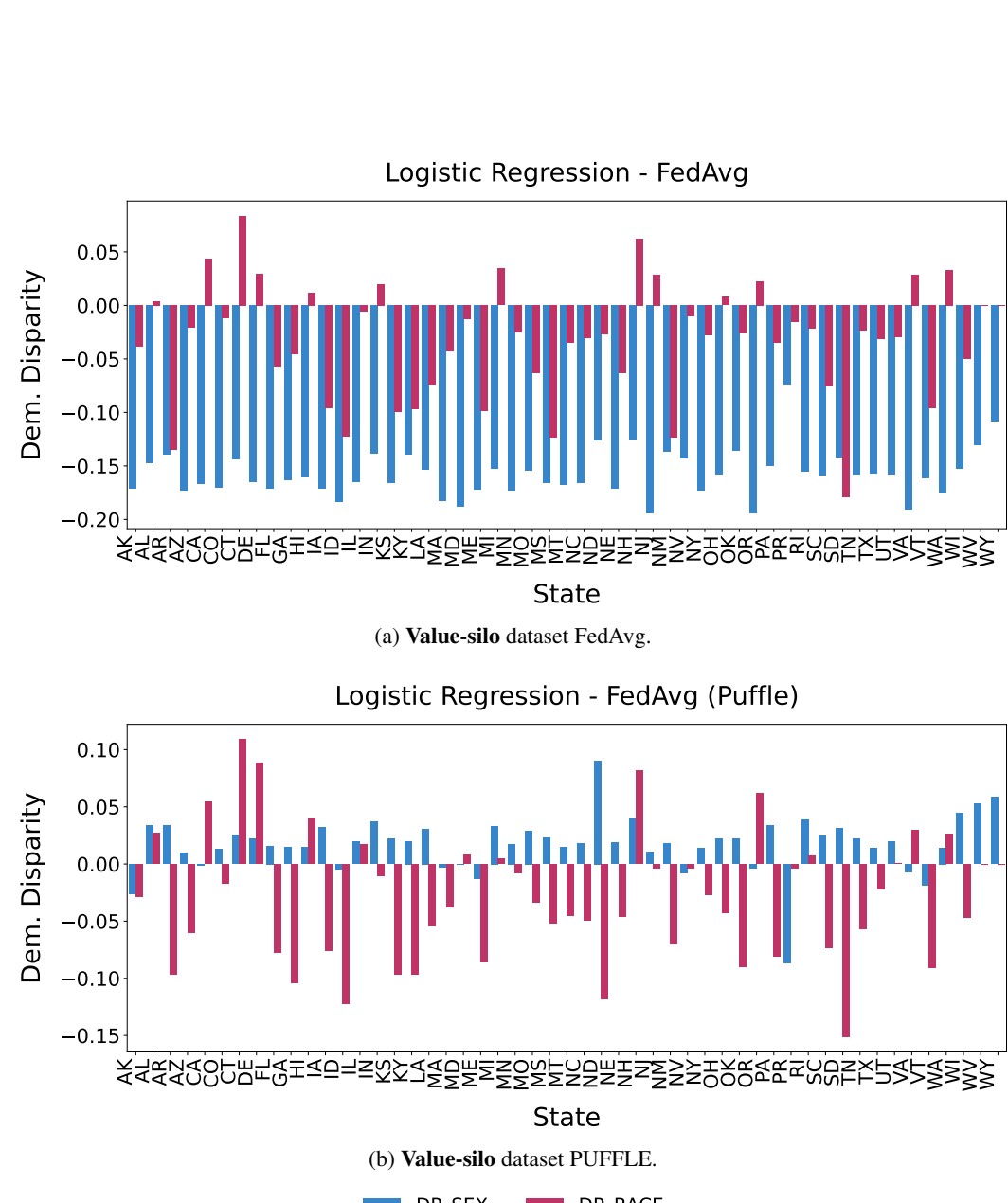

Figure 20: Individual, per attribute, bias differences in Demographic Disparity between the local Logistic Regression models vs. the FedAvg model and vs. PUFFLE for the value-silo dataset.

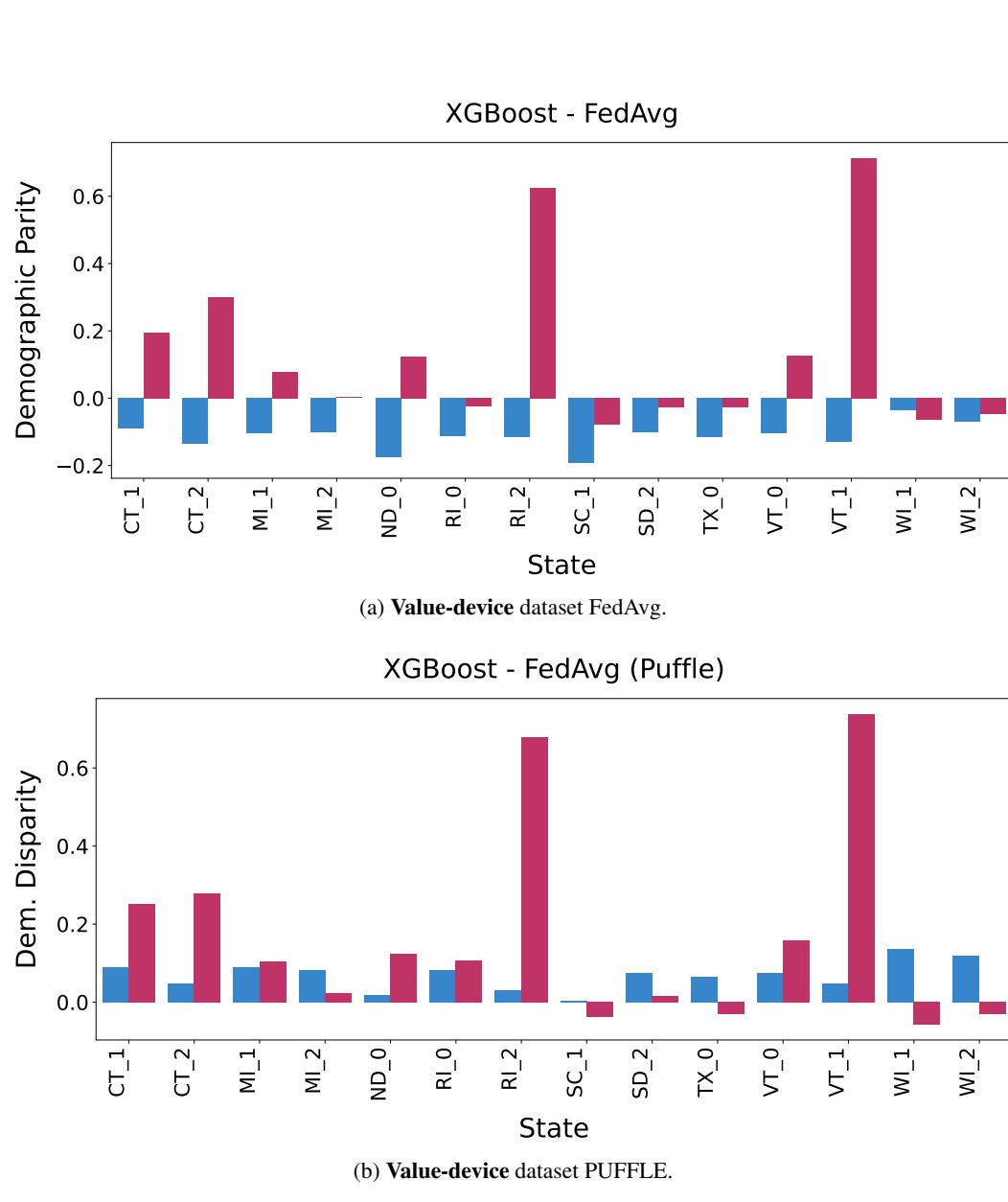

(a) **Value-device** dataset FedAvg.

(b) **Value-device** dataset PUFFLE.

Figure 21: Individual differences in Demographic Disparity between the local XGBoost models vs. the FedAvg model and vs. the PUFFLE model for the value-device dataset.

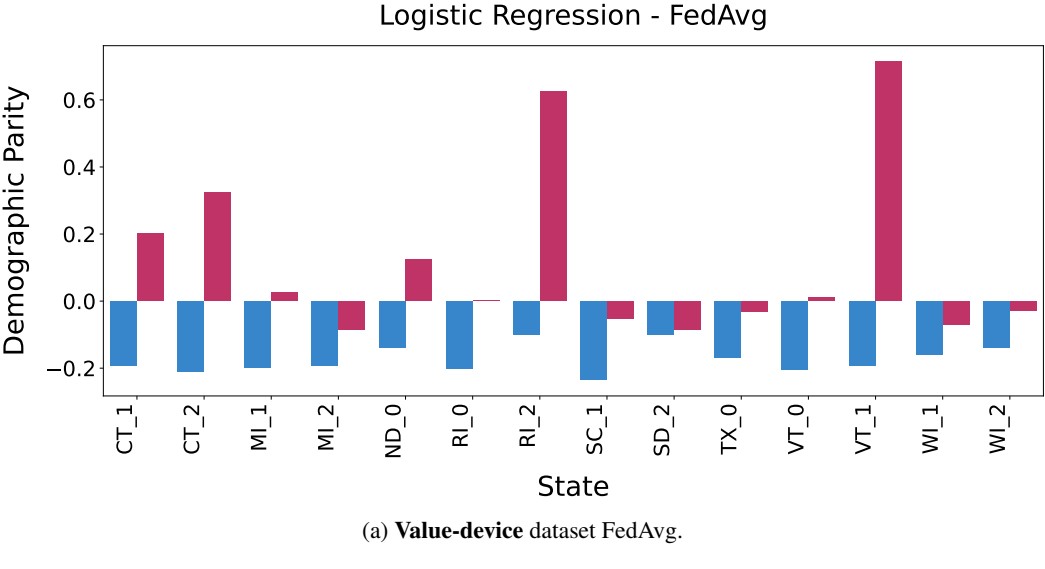

(a) **Value-device** dataset FedAvg.

(b) **Value-device** dataset PUFFLE.

Figure 22: Individual differences in Demographic Disparity between the local Logistic Regression models vs. the FedAvg model and vs. the PUFFLE model for the value-device dataset.

