# OpenReview forum: "FeDa4Fair: Client-Level Federated Datasets for Fairness Evaluation"
_ICLR.cc/2026/Conference — ICLR 2026 Conference Withdrawn Submission_

### Official Review · Reviewer_VxnW · 2025-10-25

**Soundness:** 2
**Presentation:** 2
**Contribution:** 2
**Rating:** 4
**Confidence:** 3

**Summary:**

This paper introduces the FeDa4Fair framework, which targets the challenge of insufficient standardized datasets and evaluation tools for capturing heterogeneous client-level biases in federated learning (FL) and fairness research. The authors argue that existing fair FL methods and benchmarks rely on unrealistic assumptions of uniform bias distributions across clients, failing to account for real-world heterogeneity, specifically distinguishing between value bias and attribute bias. FeDa4Fair addresses this issue by creating tabular FL datasets with controlled bias distributions, releasing benchmark datasets covering both cross-silo and cross-device settings, and providing client-level fairness evaluation. Experiments are conducted to demonstrate the performance of the proposed datasets and evaluation framework.

**Strengths:**

The paper addresses an important and under-explored problem of client-level fairness under heterogeneous bias in FL. It focuses on both value bias and attribute bias, and is well-motivated. FeDa4Fair provides a framework for generating fairness-aware FL datasets, which integrates with the Flower framework and supports both cross-silo and cross-device settings. The creation of benchmark datasets with controlled bias properties addresses the critical challenge of lacking standardized fairness-aware FL benchmarks.

**Weaknesses:**

1. FeDa4Fair primarily relies on label flipping and datapoint dropping techniques to create controlled datasets. However, the paper should provide better justification of why these methods are representative of real-world bias mechanisms, and compare or discuss how the resulting biases align with empirical biases in real FL systems.
2. The experimental validation is limited to two FL methods, which undermines FeDa4Fair’s ability to demonstrate its utility as a benchmarking tool.
3. While the fair FL field lacks consensus on benchmarks, the paper fails to compare FeDa4Fair against other related methods that address heterogeneous bias (e.g.,Federated Learning with Fair Averaging (IJCAI 2021) and mFairFL (AAAI 2024)).
4. The evaluation of cross-device settings is less rigorous than that of cross-silo settings. Although the authors mention that cross-device clients have limited data, they offer no solution for reliable fairness metric computation. The core challenge of how to measure fairness for clients with too few samples to even train a local model requires further clarification. It limits FeDa4Fair’s application to real-world cross-device FL scenarios.
5. While FeDa4Fair claims to adopt a dataset-agnostic approach, all experiments and benchmarks are based on tabular ACS data. Guidance or discussion on adapting FeDa4Fair to non-tabular modalities would help.

**Questions:**

Could the authors provide stronger justification for why FeDa4Fair’s primary bias exacerbation techniques  are representative of real-world bias mechanisms, and further compare or discuss how the resulting biases align with empirical biases in real federated learning systems? Could the authors expand evaluations to include comparisons against related methods that address heterogeneous bias? For cross-device settings, could the authors further clarify how to measure fairness for clients with too few samples to train even a local model? Additionally, could the authors add discussion on adapting the framework to non-tabular modalities to enhance its broader applicability?

---

### Official Review · Reviewer_k7gW · 2025-10-29

**Soundness:** 2
**Presentation:** 2
**Contribution:** 1
**Rating:** 2
**Confidence:** 4

**Summary:**

This paper presents FeDa4Fair, a library designed for evaluating the fairness of client-level models trained via federated learning. FeDa4Fair leverages data from ACS PUMS, offers functions to manipulate client-level data heterogeneity, and provides visualization tools to assess each client’s fairness performance in terms of Demographic Disparity (DD) and Equal Opportunity Difference (EOD). In addition, it includes four ready-to-use benchmark datasets and implements two baseline FL methods.

**Strengths:**

This work represents a valuable attempt to establish a unified framework that facilitates systematic evaluation of model fairness at multiple levels within federated learning.

**Weaknesses:**

The paper may be better suited for venues such as the NeurIPS Datasets and Benchmarks Track, given its focus on library and dataset design rather than new methodological contributions. Moreover, the current version appears narrow in scope. The work could be strengthened by expanding in the following directions:

1. Broaden dataset diversity. Although the authors claim to provide four datasets, all are derived from the same source (ACS PUMS) with different partitions or manipulations. It would be valuable to extend the framework so that the manipulation functions can be applied to more general datasets, such as COMPAS or CelebA.

2. Support general fairness metrics. At present, only DD and EOD are implemented. Allowing users to define customized metrics, and adding support for calibration-based and causal-based fairness criteria, would greatly enhance flexibility.

3. Following the above point, adding functions to support finer-grid manipulation/summarization, such as the responses conditioning on sensitive attributions would make the toolkit more comprehensive.

Minor Comments

- For cross-device datasets, increasing the number of partitions would make the setting more representative; 100 clients is not substantially larger than the 50 used in the cross-silo case.

- The phrasing “... biased towards attribute ...” was initially confusing. It appears the authors mean that the model exhibits stronger bias on one attribute than others. Rewording this helps improve clarity.

**Questions:**

Please see Weaknesses above.

---

### Official Review · Reviewer_owkt · 2025-11-01

**Soundness:** 3
**Presentation:** 3
**Contribution:** 2
**Rating:** 4
**Confidence:** 2

**Summary:**

FeDa4Fair is the first open-source library and dataset suite purpose-built to simulate, measure, and benchmark client-level fairness in federated learning (FL). It supplies four rigorously documented, heterogeneous-bias federated datasets (tabular, cross-silo & cross-device) plus an extensible Python API that lets researchers inject attribute-level or attribute-value-level bias in a controlled, reproducible way. The accompanying experiments (FedAvg vs. PUFFLE) empirically validate that global fairness scores can hide stark client-level disparities, underscoring the need for client-centric fairness evaluation.

**Strengths:**

- The authors provide interfaces for generating synthetic and semi-synthetic datasets based on real data (e.g., ACSIncome), controlling for both value and attribute bias at the client level, and supporting both cross-silo and cross-device FL setups . This covers two leading real-world bias settings often ignored by existing federated fairness tools.

- The paper’s narrative is logical, presents motivation clearly, and makes it easy to follow

**Weaknesses:**

- The paper’s contribution is largely one of data engineering: it repackages existing public datasets (ACS) with standard partitioning and bias-exacerbation techniques to produce client-level federated benchmarks. While the need to measure fairness per client is duly highlighted, the idea itself is neither theoretically elaborated nor algorithmically advanced here.
- No new fairness-aware FL algorithms, optimization protocols, or metrics are introduced; the intellectual effort is confined to curating and documenting the data splits. The resulting benchmarks are restricted to demographic prediction tasks (income/employment) derived from ACS, leaving their relevance to other domains, modalities, or task formulations unexplored.
- Extensive experiments mostly re-confirm well-documented observations—e.g., that FL can amplify bias and that fairness interventions hurt accuracy—without yielding fresh insight or actionable guidance. To move beyond a useful but incremental resource, the work should supply methodological innovations (novel aggregation schemes, client-specific fairness constraints, or theoretically grounded trade-off analyses) rather than dataset packaging alone.

**Questions:**

See weakness

---

### Note · Authors · 2025-12-04

**Comment:**

We thank all reviewers for their helpful reviews. However, we decided to continue working more on this paper.

**Withdrawal Confirmation:**

I have read and agree with the venue's withdrawal policy on behalf of myself and my co-authors.